# 🐧 Pengi: An Audio Language Model for Audio Tasks

**Soham Deshmukh**[1]    **Benjamin Elizalde**[1]    **Rita Singh**[2]    **Huaming Wang**[1]

[1]Microsoft    [2]Carnegie Mellon University

{sdeshmukh, benjaminm, huawang}@microsoft.com, rsingh@cs.cmu.edu

## Abstract

In the domain of audio processing, Transfer Learning has facilitated the rise of Self-Supervised Learning and Zero-Shot Learning techniques. These approaches have led to the development of versatile models capable of tackling a wide array of tasks, while delivering state-of-the-art performance. However, current models inherently lack the capacity to produce the requisite language for open-ended tasks, such as Audio Captioning or Audio Question Answering. We introduce Pengi, a novel Audio Language Model that leverages Transfer Learning by framing all audio tasks as text-generation tasks. It takes as input, an audio recording, and text, and generates free-form text as output. The input audio is represented as a sequence of continuous embeddings by an audio encoder. A text encoder does the same for the corresponding text input. Both sequences are combined as a prefix to prompt a pre-trained frozen language model. The unified architecture of Pengi enables open-ended tasks and close-ended tasks without any additional fine-tuning or task-specific extensions. When evaluated on 21 downstream tasks, our approach yields state-of-the-art performance in several of them. Our results show that connecting language models with audio models is a major step towards general-purpose audio understanding [1].

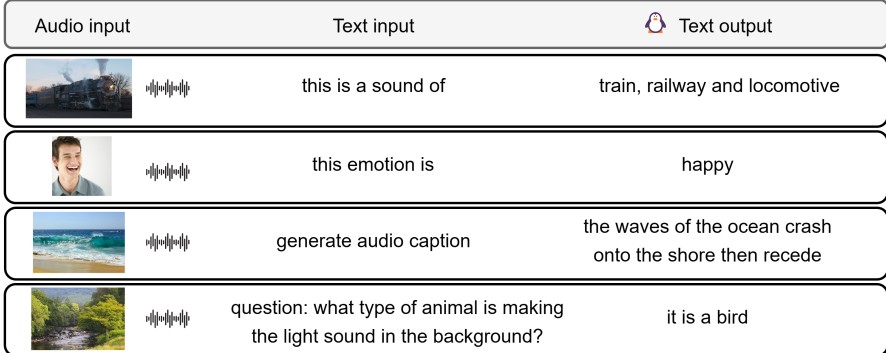

Figure 1: Examples of audio and text prompt inputs and their corresponding textual responses. Images are for illustration purposes only. Our proposed model Pengi enables close-ended tasks, such as classification or retrieval and open-ended tasks, such as captioning or question & answering.

## 1  Introduction

Machine Listening breaks down audio understanding into separate and independent audio tasks. For example, Sound Event and Scene Classification, Audio Retrieval, and Audio Captioning. Because these audio tasks are intrinsically related, we can leverage from Transfer Learning (TL). TL focuses on applying knowledge gained while solving one task to solve a related task. The learning method involves pre-training a model with a large compilation of datasets from different tasks followed by fine-tuning on a target dataset. These models have shown the potential to learn general-purpose audio

---

[1]Code is available here: https://github.com/microsoft/Pengi

37th Conference on Neural Information Processing Systems (NeurIPS 2023).

representations [53] that can successfully be used in a variety of downstream tasks. To leverage from larger amounts of audio that is unlabeled, the community has employed Self-Supervised and Unsupervised Learning [50, 51, 44, 19]. These methods do not require labels [53, 7] and have achieved state-of-the-art performance. However, both methods require an additional fine-tuning step before they can be applied to any downstream task.

To address this drawback, another Transfer Learning (TL) method called Zero-Shot Learning provides direct inference capabilities and removes the need of fine-tuning. These models use contrastive objectives to learn the similarity between natural language descriptions and audio content to provide a score that identifies the most probable class label for a given testing audio. Examples are CLAP [15], Mulan [26], and LAION-CLAP [58]. Despite not seeing the training data of a target task, Zero-Shot models achieve surprising performance in close-ended tasks, such as classification and retrieval. However, these models inherently lack the capacity to produce the requisite language for open-ended tasks, such as Audio Captioning or Audio Question Answering (AQA).

Current audio models that can perform open-ended tasks do not support or have not been evaluated on closed-ended tasks [37, 31]. It is yet to be explored how to leverage TL to enable both types of tasks in the audio domain. We drew inspiration from recent advances in Natural Language Processing (NLP) and Visual Language Models (VLM). In NLP, Raffel et. al. [49] explored a unified framework called T5 where all text-based tasks are framed as text input to text output problems. T5 was trained with a single objective function and supported a diverse set of tasks, like translation, question & answering, and classification. FLAN [8] showed that language models trained on a collection of text tasks phrased as instructions, enabled models to respond better to similar instructions at inference time. This TL technique showed performance improvement across a range of models, prompting setups, and evaluation tasks. On the other hand, VLM incorporates visual information by combining a language model and an image encoder to transfer knowledge across modalities. Tasks are framed as text and image input to text output problems. Captioning training consists of optimizing a text generation objective, and can transfer moderately well to visual question & answering in the zero-shot settings. Examples include, Frozen [52], Flamingo [2], and other models [54, 52, 2, 42, 39]. But their performance on close-ended tasks still lags behind contrastive models [47, 59]. In the audio domain, there are no models that resemble any of these capabilities, let alone that support both close-ended and open-ended audio tasks simultaneously.

In this paper, we introduce Pengi, a novel Audio Language Model (ALM) that takes as input, an audio recording and a text prompt, and generates free-form text as output. To the best of our knowledge, the following contributions are achieved for the first time in the literature:

- A novel Audio Language Model capable of supporting multiple close-ended and open-ended audio tasks without any additional fine-tuning or task-specific extensions of the architecture. Pengi draws inspiration from VLM but tackles intrinsic challenges in the audio domain.

- We propose a new learning framework where we frame all audio tasks as audio and text input to text output tasks. Our framework uses a single training procedure and a captioning objective function. For training, we designed new audio task templates inspired by Instruction Tuning.

- We extensively evaluated Pengi on 21 downstream tasks across various audio domains yielding state-of-the-art performance in several of them. Thus, establishing a baseline for general-purpose ALM.

## 2   Related Work

**Audio Language Models.** In the domain of audio processing, Transfer Learning has facilitated the rise of Self-Supervised Learning and Zero-Shot Learning techniques [50, 51, 43, 22, 25, 24, 3, 15, 26, 23, 57, 58, 41, 12, 14, 16]. These approaches have led to the development of versatile models capable of tackling a wide array of tasks, while delivering SoTA performance. However, current models can tackle either close-ended tasks or open-ended tasks. ALM pose a new learning paradigm for audio processing that can support all tasks. The language modeling approaches to audio find utility in generating audio given an input description [4, 1]. But it is yet to be explored how to train them for general-purpose audio understanding and what their performance would be.

**Language Models**. Transfer Learning has been extensively utilized in Natural Language Processing with the recent shift to Zero-Shot and Few-Shot Learning [29, 48, 5, 55]. The work by Raffel et. al. [49] explored a unified framework for text tasks by converting all text-based tasks into the text-to-text

format. The experimental results showed the methods can achieve SoTA results when combined and scaled. FLAN [55] released in 2022 uses instruction fine-tuning to fine-tune an existing language model on a large set of varied instructions. Pengi adapts a similar idea for the audio domain, where each audio-tasks is considered a text generation task conditional on the input text and input audio. This allows audio tasks to be represented in (audio-text)-text format and enables learning a single unified model for all the tasks. For training, we created (audio-text)-text templates for audio tasks and trained Pengi with them.

**Visual Language Models.** Inspired by the success of Transfer Learning and Few-Shot Learning in NLP, a host of VLM were proposed for vision tasks. VLM intend to extend the pre-trained language model and adapt them to incorporate visual information. VisualBERT [35] and SimVLM [54] explored different ways to convert images into tokens and jointly train the model on interleaved images and text. Inspired by prefix-tuning [36] and prompt-tuning [34], Frozen [52] and Clipcap [42], use a frozen language model and align the image embeddings for the language model. To better fuse image information, Flamingo [2] uses a gated-cross-attention dense layer in the language model. The interleaved image-text training also enables Flamingo to do few-shot learning. Drawing parallels with VLM, Pengi can be considered an ALM based on *audio conditional prefix tuning* where the prompt is produced by an audio encoder.

# 3  Approach

In this section, we describe Pengi, a novel Audio Language Model that leverages Transfer Learning by framing all audio tasks as text generation tasks. It takes as input, an audio recording and a text prompt, and generates free-form text as output. The unified architecture in Figure 2 enables open-ended tasks and close-ended tasks without any additional fine-tuning or task-specific extensions of the architecture.

## 3.1  Unified Architecture

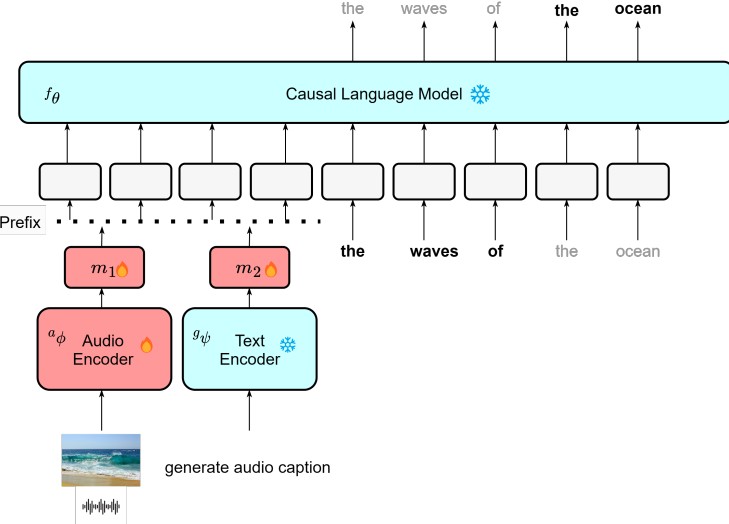

Figure 2: 🐧 Pengi has a unified architecture that takes as input, an audio recording and a text prompt, and generates free-form text as output. At training, the architecture learns an audio encoder $a_\phi$ and a mapping network $m_1$ to represent an input audio as a sequence of continuous embeddings. A frozen text encoder $g_\psi$ and a learnable mapping $m_2$ do the same for the corresponding text input. Both sequences are concatenated as a prefix to leverage from a pre-trained frozen autoregressive language model $f_\theta$ to perform multiple tasks. At inference, the language model generates tokens autoregressively conditioned on the audio and text input.

**Audio Encoder.** The audio encoder $a_\phi$ transforms the raw audio input into an audio embedding. We used the audio transformer backbone from CLAP [15] as our audio encoder due to its success in diverse audio and multimodal tasks. Models in Computer Vision [42, 2, 39] use a frozen image encoder like CLIP, but CLAP is trained on a magnitude smaller collection of audio-text pairs. Therefore, we unfroze its weights for our training procedure.

**Text Encoder.** The text encoder $g_\psi$ transforms the input text prompt into a text embedding. The prompt can be any form of natural language, such as a task-specific prompt or a question. The text encoder is frozen so its weights are not updated during training. The text encoder can be any off-the-shelf text encoder and allows our architecture to learn and perform well in close-ended tasks.

**Mapping Networks and Prefix.** To construct the prefix to be fed to the causal language model, we used two mapping networks ($m_1$ and $m_2$). The mapping networks [42] convert an embedding into a sequence of $k$ embeddings. The audio embedding is transformed by $m_1$ and the text embedding by $m_2$, both are trainable. Both sequences are concatenated to form the fixed-length prefix.

**Causal Language Model.** To generate the text output we used a pre-trained autoregressive causal language model which is kept frozen during training and inference [52]. Even though the language model is frozen, the audio prefix receives gradients enabling the parameters of mapping network ($m_1$) and audio encoder $a_\phi$ to be optimized with gradient descent and backpropagation. At inference, the language model generates tokens autoregressively conditioned on the audio and text prefix.

### 3.2 Training and Inference

We propose a new learning framework where we frame all audio tasks as audio and text input to text output tasks. Our framework uses a single training procedure and objective function. Let the training data in audio-text-to-text format be referred to as $\{x^i, t^i, c^i\}$ where $x^i$, $t^i$ and $c^i$ are the $i^{th}$ audio file, $i^{th}$ input text, and $i^{th}$ output text or caption respectively.

To create a prefix, the audio encoder $a_\phi$ and mapping network $m_1$ projects the audio $x^i$ into a sequence of $k$ embeddings. Similarly, the text encoder $g_\psi$ and mapping network $m_2$ projects the input text $t^i$ into a sequence of $k$ embeddings. Both sequences are concatenated to form prefix $p^i$ for the pre-trained frozen language model $f_\theta$.

$$p^i = p_1^i, ..., p_{2k}^i = \text{concat}\{m_1(a_\phi(x^i)), m_2(g_\psi(t^i))\} \tag{1}$$

The language model $f_\theta$ is fed with the prefix-caption concatenation of all $\{z_i\}_{i=1}^N$, where $z_i$ is:

$$z^i = p_1^i, ..., p_{2k}^i, c_1^i, ..., c_l^i \tag{2}$$

The model is trained as a standard captioning system, where it learns to predict a caption (text tokens) $c^i$ conditioned on the prefix in an autoregressive fashion. We used Cross-Entropy as the loss function:

$$\mathcal{L} = -\sum_{i=1}^N \sum_{j=1}^l \log p_\gamma(c_j^i | p_1^i, ..., p_{2k}^i, c_1^i, ..., c_{j-1}^i) \tag{3}$$

where $\gamma$ denotes model's trainable parameters which include audio encoder parameters $\phi$ and parameters from both mapping networks. The text encoder and the causal language model are frozen.

At inference time, the prefix is constructed using the test audio and a text prompt. The causal language model $f_\theta$ generates the next token sequentially conditioned on the prefix. The language model assigns probabilities to all vocabulary tokens at each prediction, which are used to determine the next token depending on the choice of decoding. In our experiments, we used beam search decoding with a beam size of 5 for inference and downstream tasks.

## 4 Experiments

### 4.1 Training Datasets and Templates

Our Audio Language Model Pengi is trained on a collection of audio-text tasks phrased as instruction templates. The templates are inspired by instruction tuning and enable models to respond better to similar instructions at inference time. This TL technique is novel for audio and yielded performance improvement across a range of input prompting examples and downstream tasks.

The training datasets are modified to adapt to our proposed framework (audio-text)-to-text format by constructing 8 audio-task templates. Before our study, there was no evidence that the templates could lead to good performance across open- and close-ended tasks. Each template consists of audio input, input text prompt, and text output. Examples are "this is the sound of", "this emotion is" or "question: {question}". All the templates are in Table 1, out of which one template is the Auxiliary task "generate metadata". With it, we add audio-text pairs that are not task-specific. Drawing parallels, this training data setup is inspired by instruction tuning format of FLAN [55, 8]. Defining new templates or variations of the ones proposed here is a promising direction to explore.

| Task | Input prompt | Output format | | Task | Input prompt | Output format |
|---|---|---|---|---|---|---|
| Audio Captioning | generate audio caption | {caption} | | Speech Emotion Recognition | this emotion is | {emotion} |
| Audio QA | question: {question} | {answer} | | Speech Sentiment Recognition | this sentiment is | {sentiment} |
| Sound Event Classification | this is a sound of | {event a}, {event b}, .. | | Music Analysis | music analysis | this is a sound of music in language {language} and genre {genre} .. |
| Acoustic Scene Classification | this acoustic scene is | {scene} | | Music Note Analysis | this music note is | produced by {instrument}, pitch {pitch}, .. |
| | | | | Auxiliary | generate metadata | {metadata} |

Table 1: The training datasets are modified to adapt to our proposed framework (audio-text)-to-text format by constructing 8 audio-task templates. Each template consists of audio input, input text prompt, and text output. The {} symbol indicates variable content. The Auxiliary task template allowed us to add audio-text pairs that are not task-specific.

The training data is collected from multiple audio datasets coming from different sources. In all, we collected 3.4 million audio-text pairs and mapped them to the 8 templates. The number of training pairs makes this model one of the largest if not the largest non-speech audio model in literature. We use only the training set of each dataset. The datasets and their mapping to a task are the following. Sound Event Classification: AudioSet [21], FSD50K[20]; Acoustic Scene Classification: CochlScene [27]; Speech Emotion and Sentiment Recognition: MSP Podcast [38], CMU MOSI [60], CMU MOSEI [61], MELD [46]; Music Analysis: NSynth [17], FMA [9]; Audio Captioning: AudioCaps [30], ClothoV2 [13]; Audio Question and Answering: ClothoAQA [37]; Auxiliary: WavText5K [11], SoundDescs [33], MACS [40], WavCaps [41], FreeSound [18] and FindSound[2].

## 4.2 Downstream Tasks

The unified architecture of Pengi enables open-ended tasks and close-ended tasks.

**Open-ended tasks.** This task type requires free-form text generation and there is flexibility in the correctness of the output. Examples are Audio Captioning and AQ&A. Pengi will take as input the testing audio and the desired prompt to generate the text output. It does not require any additional fine-tuning or task-specific components.

**Close-ended tasks.** This task type is restricted to predefined values that can be classes or numbers. Examples are classification and retrieval. Pengi will take as input the testing audio and the desired prompt. Ideally, the free-form text output from Pengi should contain the exact predefined value. For example, a predefined class is "dog" but Pengi may output "dog barking" or "canine". Although these answers are reasonable, they are incorrect under most metrics. To evaluate the correctness, we proposed two methods: Log-likelihood and Text matching (Fig. 3). Unless explicitly mentioned, all experiments in our paper use the Text-matching method for evaluation.

*Log-likelihood:* We take the concatenated prefix from a testing audio, the prompt, and append one of the predefined values (e.g class name, number) to create a candidate output. We would have $N$ candidate outputs corresponding to $N$ predefined values. For example in classification, if we have 100 testing audios and 5 classes, we would have 5 output candidates per audio. The outputs and the predefined values are used to compute Log-likelihood scores and determine the model's prediction. This method is expensive for the extensive evaluation in our study.

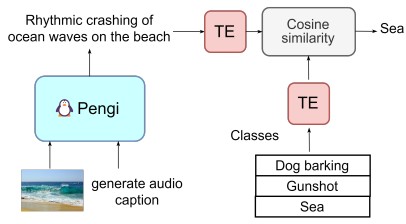

Figure 3: Text-matching method used during inference for close-ended tasks. TE indicates Text Embedding.

*Text-matching:* In this setup, the free-form output is matched to the predefined values using text embeddings (Fig.3). For example, in a classification setting, we compute sentence-level text embeddings for Pengi's output and for all the class labels in a given dataset. Then, we calculate cosine similarity to determine the model's prediction. We used Pengi's text encoder to compute the embeddings, but any off-the-shelf text encoder could be used.

---

[2]https://www.findsounds.com

**Downstream tasks.** We used 21 downstream tasks (Table 2) to benchmark the open-ended and close-ended capabilities of Pengi. The open-ended tasks consist of Audio Captioning and AQA. The close-ended tasks consist of classification, regression, and retrieval. Datasets like Clotho have more than one type of annotations, so they are used for multiple tasks like Audio Captioning and Text-to-Audio Retrieval.

| Domain | Dataset | Files | Dur. (secs) | Output Type | Metric | Setup |
|---|---|---|---|---|---|---|
| Audio Captioning | Clotho | 7k | 15 - 30 | Cap. | SPIDEr | train/val/test |
| | AudioCaps | 39k | 10 | Cap. | SPIDEr | train/val/test |
| Audio Question Answering | ClothoAQA | 2k | 15 - 30 | Q&A | ACC | train/val/test |
| Sound Event Classification | ESC50 | 2k | 5 | MC (50) | ACC | 5 folds |
| | FSD50K | 51k | 0.3 - 30 | ML (200) | mAP | train/val/test |
| | UrbanSound8K | 8k | ≤ 4 | MC (10) | ACC | 10 folds |
| | DCASE2017 Task4 | 52k | 10 | MC (17) | ACC | train/val/test |
| Music Analysis | GT. Music Speech | 120 | 30 | B (2) | ACC | 10 folds |
| | GT. Music Genre | 1k | 30 | MC (10) | ACC | 10 folds |
| Instrument Classification | Beijing Opera | 236 | 4.77 | MC (4) | ACC | 5 folds |
| | NS. Instruments | 305k | 4 | MC (11) | ACC | train/val/test |
| Music Note Analysis | NS. Pitch | 305k | 4 | Reg. | ACC | train/val/test |
| | NS. Velocity | 305k | 4 | MC (11) | ACC | train/val/test |
| | NS. Sonic | 305k | 4 | ML (10) | ACC | train/val/test |
| Acoustic Scene Classification | TUT 2017 | 6.3k | 10 | MC (15) | ACC | train/val/test |
| Emotion Recognition | CREMA-D | 7k | 5 | MC (6) | ACC | 5 folds |
| | RAVDESS | 2.5k | ≤ 5 | MC (8) | ACC | 5 folds |
| Vocal Sound Classification | Vocal Sound | 21k | 5 | MC (6) | ACC | train/val/test |
| Surveillance | Surveil. Applications | 585 | ≤ 33 | MC (6) | ACC | train/val/test |
| Text-to-Audio Retrieval | Clotho | 7k | 15 - 30 | Ret. | R@1 | train/val/test |
| | AudioCaps | 39k | 10 | Ret. | R@1 | train/val/test |

Table 2: We extensively evaluated Pengi across 21 downstream tasks from various domains. The first two domains are open-ended tasks and the rest are close-ended tasks. For the "Output Type" column, Cap. refers to captioning, MC to multiclass, B indicates binary, Reg. indicates regression, and Ret. retrieval.

### 4.3 Implementation details

**Encoders and mappers.** We used the audio transformer HTSAT[6] as our audio encoder and CLIP's [47] text encoder. The audio is sampled at 44.1 kHz and is converted to a log Mel spectrograms with 64 Mel bins, a hop size of 320 ms, and a window size of 1024 ms in the range of 50-8000 Hz. We randomly truncated all audio files to 7 seconds in length for HTSAT. The max length of the text encoder is set to 40 for computational efficiency. We performed another step of CLAP (Contrastive Language-Audio Pretraining) training using the above two encoders [15]. This enables experiments where the audio encoder can be kept frozen to see the utility of CLAP's [15] audio embeddings similar to VLM [42, 52, 2]. The mapping networks $m_1$ and $m_2$ each use an 8-layer transformer with a prefix length of 40. The total prefix length after concatenating the audio and text is 80. The hyper-parameters of the encoders and the CLAP training are mostly left as in the original papers, the details are in Appendix D.

**Causal Language Model.** We used the GPT2 line of models, specifically GPT2-base (124M). The model is kept frozen through all the experiments.

**Pre-training.** We used Adam Optimiser [32] for 60 epochs and with a batch size of 384 on 20 V100 GPUs. We used a linear schedule with 2000 warmup steps and a base learning rate of 1e-4.

## 5 Results

### 5.1 Benchmarking Pengi

We assessed Pengi on 21 downstream tasks covering various domains. Pengi is the first audio model that can perform both, open-ended and close-ended tasks. A fair comparison against another model that can perform both is not possible. We chose CLAP [15] as the baseline because it is the only Zero-Shot model with a comprehensive evaluation (16 downstream tasks). The next best evaluation was only on 8 tasks. Thus, providing no evidence of performance across domains like speech and music, which tend to be the most difficult. Moreover, we compared against SoTA results even if it came from different models and learning methods. We compared against SoTa Zero-Shot models in Table 8, a subset of Table 3, for Sound Event Classification. Even against SoTA from supervised

| | Audio Captioning ↑ | | AQA ↑ | Sound Event Classification ↑ | | | |
|---|---|---|---|---|---|---|---|
| Model | AudioCaps | Clotho | ClothoAQA | ESC50 | FSD50K | US8K | DCASE17 Task 4 |
| CLAP | ✗ | ✗ | ✗ | 0.826 | 0.3024 | **0.7324** | 0.3 |
| Pengi | **0.4667** | **0.2709** | **0.6453** | **0.9195** | **0.4676** | 0.7185 | **0.338** |

| | Acoustic Scene Classification↑ | Music ↑ | | Instrument Classification ↑ | | Music Note Analysis↑ | | |
|---|---|---|---|---|---|---|---|---|
| Model | TUT2017 | Music Speech | Music Genres | Beijing Opera | Instrument family | NS. Pitch | NS. Velocity | NS. Qualities |
| CLAP | 0.2963 | **1.0** | 0.252 | 0.2963 | 0.2949 | - | - | - |
| Pengi | **0.3525** | 0.9688 | **0.3525** | **0.6229** | **0.5007** | **0.8676** | **0.3728** | **0.386** |

| | Emotion Recognition↑ | | Vocal Sound Classification↑ | Action Recog.↑ | Surveillance.↑ |
|---|---|---|---|---|---|
| Model | CREMA-D | RAVDESS | Vocal Sound | ESC50 Actions | SESA |
| CLAP | 0.1784 | 0.1599 | 0.4945 | 0.497 | **0.7487** |
| Pengi | **0.1846** | **0.2032** | **0.6035** | **0.5277** | 0.5402 |

Table 3: We used CLAP [15] as a baseline comparison because of its strong performance on a wide range of downstream tasks. The '-' symbol indicates numbers were not available, whole '✗' indicates that the model cannot support the task. Higher is better for all numbers. The evaluation metric is mAP for FSD50k, AudioSet, ESC50-Actions, and NSynth sonic; F1 score for DCASE17; and SPIDEr for AudioCaps and Clotho captioning. All other downstream tasks use Accuracy.

learning models in Tables 5 and 7 for AQ&A and Audio Captioning respectively. Table 9, against SSL, supervised and trained on speech audio models.

**Open-ended tasks.** Pengi sets new state-of-the-art performance for open-ended tasks. We used Audio Captioning and AQA for open-ended tasks. The CLAP model can only support close-ended tasks and cannot perform open-ended tasks without additional modules and fine-tuning. Therefore, we compared against supervised trained models in Section 5.2.

**Close-ended tasks.** Pengi performs better than CLAP on most audio classification tasks, and can also outperform the literature. Although CLAP and Pengi employed different learning methods and used a different amount of training data, it is to be noted that Pengi can compete with strong contrastive methods like CLAP and other methods in the literature.

## 5.2 Audio Captioning and AQA

**Audio Captioning.** Pengi's performance outperformed supervised models in the two captioning tasks AudioCaps and Clotho, as shown in Table 7. The captioning competition IEEE DCASE 2022 [3] ranks models based on the metric SPIDEr, a combination of CIDEr and SPICE. Specifically, for AudioCaps Pengi outperformed the literature by a relative 6.6% and for Clotho by a relative 26%. All models used both, AudioCaps and Clotho datasets in training. One of the best captioning models is from Kim et al. [31]. The authors followed a similar training procedure to ours with audio encoders and a language model. Unlike Pengi, which uses a single audio encoder, they employed two mapping networks to capture both global and temporal features from the audio. Despite having two audio representations, the model underperformed our approach.

Similar to Multi-Task Learning [62, 10], we hypothesize that learning a shared audio encoder and mapping networks helps Pengi to solve individual tasks better. We addressed this hypothesis by conducting an ablation study in Table 4. In experiment *A*, we trained and evaluated Pengi only on audio-captioning data with text prompts of "generate audio caption". Then, we contrasted audio captioning performance against experiment *B*, where we trained on data across different tasks, in other words, our proposed setup in this paper. From Table 4, we see consistent improvement in both AudioCaps and Clotho downstream tasks. Specifically, experiment *B* outperforms experiment *A* by a relative 2.5% and 2.3% on AudioCaps and Clotho respectively. This indicates that Pengi's shared architecture does help in improving performance on individual tasks.

| Exp. | Eval. dataset | Audio Captioning ↑ | |
|---|---|---|---|
| | | BLUE$_1$ | SPIDEr |
| A | AudioCaps | 0.6439 | 0.4551 |
| B | AudioCaps | **0.6912** | **0.4667** |
| A | Clotho | 0.5619 | 0.2648 |
| B | Clotho | **0.5702** | **0.2709** |

Table 4: Effect of shared audio encoder training

| Model | Audio Q&A ↑ |
|---|---|
| | Acc |
| M$_1$ | 0.575 |
| M$_2$ | 0.627 |
| M$_3$ | 0.635 |
| Pengi | **0.645** |

Table 5: AQ&A results

| Model | Retr. | Text-to-Audio Retrieval ↑ | | |
|---|---|---|---|---|
| | | R@1 | R@5 | R@10 |
| Chen et al. | Clotho | 1.5 | 4.4 | 7.5 |
| Gont. et al. | Clotho | 2.1 | 7.0 | 12.0 |
| Mei et al. | Clotho | 4.0 | 14.1 | 21.6 |
| Kim et al. | Clotho | 7.6 | 19.6 | 28.8 |
| Soham et al. | Clotho | 16.7 | 41.0 | 54.1 |
| Pengi | Clotho | **9.4** | **26.1** | **36.7** |

Table 6: T2A Retrieval results

[3] https://dcase.community/challenge2022/task-automatic-audio-captioning

| Model | Eval. dataset | BLUE$_1$ | BLUE$_2$ | BLUE$_3$ | BLUE$_4$ | METEOR | ROUGE$_L$ | CIDEr | SPICE | SPIDEr |
|---|---|---|---|---|---|---|---|---|---|---|
| Chen et al. | AudioCaps | 0.489 | 0.292 | 0.178 | 0.106 | 0.152 | 0.346 | 0.265 | 0.093 | 0.179 |
| Gontier et al. | AudioCaps | 0.635 | 0.461 | 0.322 | 0.219 | 0.208 | 0.450 | 0.612 | 0.153 | 0.383 |
| Mei et al. | AudioCaps | 0.682 | 0.507 | 0.369 | 0.266 | 0.238 | 0.488 | 0.701 | 0.166 | 0.434 |
| Kim et al. | AudioCaps | **0.708** | **0.547** | **0.402** | **0.283** | **0.238** | **0.499** | 0.710 | 0.167 | 0.438 |
| Pengi | AudioCaps | 0.691 | 0.419 | 0.371 | 0.253 | 0.232 | 0.482 | **0.752** | **0.182** | **0.467** |
| Chen et al. | Clotho | 0.516 | 0.325 | 0.215 | 0.141 | 0.153 | 0.350 | 0.314 | 0.102 | 0.208 |
| Gontier et al. | Clotho | 0.461 | 0.282 | 0.182 | 0.117 | 0.136 | 0.318 | 0.251 | 0.083 | 0.167 |
| Mei et al. | Clotho | 0.516 | 0.318 | 0.204 | 0.127 | 0.157 | 0.351 | 0.313 | 0.105 | 0.209 |
| Kim et al. | Clotho | 0.539 | 0.346 | 0.227 | 0.142 | 0.159 | 0.366 | 0.319 | 0.111 | 0.215 |
| Pengi | Clotho | **0.57** | **0.369** | **0.242** | **0.15** | **0.172** | **0.375** | **0.416** | **0.126** | **0.271** |

Table 7: Pengi outperforms the best Audio Captioning performance from supervised models. All models used both, AudioCaps and Clotho datasets in training. SPIDEr is the metric used to rank models in IEEE DCASE Challenge. Higher is better for all metrics.

**AQA.** Pengi outperformed the existing literature [37]. Authors in [37] collected the only dataset available (ClothoAQA). They converted the AQA task into a classification task, instead of a generation task. Authors trained and fine-tuned a model in a supervised setup. In contrast, we used the free-form text from Pengi, where the answer is correct only when it directly matches the human response. Note that Pengi includes the training set of ClothoAQA among its training sets, but there is no further fine-tuning on this task. The results are shown in Table 5. The first column indicates three different baseline models from [37]. Pengi achieved 64.5% and outperformed the existing supervised benchmark by a relative 1.5%.

## 5.3 Zero-Shot Sound Event Classification

We compared Pengi's classification performance against Zero-Shot contrastive models in the literature. The existing literature restricts the training and evaluation tasks to a few sound event datasets. Hence, we matched our comparisons to sound event datasets. The downstream datasets of ESC50, US8k, DCASE17 Task4 contain audio files and labels not seen by Pengi during training. We considered these three datasets to constitute a zero-shot setup for Pengi. For FSD50k, the audio files in the training split have been used for training Pengi. Hence, we do not consider this a pure zero-shot setup but nonetheless, report numbers for insights.

On Zero-Shot ESC50 performance, Pengi beats AudioCLIP [23], CLAP [15], and LAION CLAP [58] by 32%, 11%, and 1% respectively (See Table 8). Interestingly, human performance on ESC50 is 81% accuracy and Pengi's performance is 92%. Mei et. al. [41] added ChatGPT augmented audio-text pairs to CLAP training [58] and showed an improvement in performance from 91% to 94% on ESC50. On US8k, Pengi performed better than Wav2CLIP and AudioCLIP but lower than CLAP and LAION CLAP. Overall, even though Pengi is a text generation model, its Zero-Shot performance on close-ended Sound Event Classification is competitive.

| Model | Zero-Shot Sound Event Classification ↑ | | | |
|---|---|---|---|---|
| | ESC50 | FSD50K | US8K | DCASE17 Task 4 |
| Wav2CLIP | 0.414 | 0.030 | 0.404 | - |
| AudioCLIP | 0.694 | - | 0.653 | - |
| CLAP | 0.826 | 0.302 | 0.732 | 0.3 |
| LAION | 0.91 | - | **0.77** | - |
| Pengi | **0.92** | **0.468** | 0.719 | **0.338** |

Table 8: The literature on Zero-Shot audio models only reports performance on Sound Event Classification datasets. Pengi's classification performance is competitive. The '-' indicates numbers are not available. The evaluation metric for DCASE17 is the F1 score while FSD50K employs mAP, ESC50 and US8K use Accuracy.

## 5.4 Text-to-Audio Retrieval

For Text-to-Audio Retrieval in a contrastive learning setup, the user query is converted into a text embedding which is then used to retrieve the top $k$ audios by their audio embeddings [11, 58]. Pengi is a generative model and does not allow a contrastive setup. Although Pengi has an audio encoder and a text encoder that could replicate the contrastive setup, we wanted to evaluate our model from the generative perspective. First, Pengi is used to index a database by generating audio captions for all the audio recordings. Second, the user text query is matched directly to the dataset captions. The associated audio files of the top $k$ dataset captions are considered to be the top $k$ retrieved audio. Note that the cosine similarity computation is between two text embeddings and not audio and text embeddings. Thus, the quality of generated captions for indexing the dataset is important for retrieval performance.

In Table 6, we compared Pengi's Text-to-Audio retrieval performance against the literature. The models used for comparison are audio captioning models using the above-described procedure of indexing and query matching, and not the contrastive-like setup. Pengi outperforms the literature on R@1. However, contrastive models [15],[58], [11] are substantially better than generative models for the task of directly matching text to audio for retrieval. An example of contrastive model performance is shown in Table 6 as a gray row.

### 5.5 Next text-token prediction for learning audio representations

Pengi uses next-text token prediction to learn audio representations, hence a natural question is: *"Can next text-token prediction objective help in learning general purpose audio representations?"*. To answer this question, we performed linear probe [47] and shallow learning [53] experiments. After Pengi's pre-training, we took the audio encoder $a_\phi$ in Fig 2 and trained one, two, or three fully-connected linear layer(s) with cross-entropy on top. Note that, we kept Pengi's audio encoder frozen and it did not include the mapping network $m_1$. We selected representative datasets from the domain of Sound Events, Music, and Speech Emotion for the linear probe experiment. Pengi's linear probe (one layer) and shallow learning (two or three layers) numbers are compared against the best single model submissions from the HEAR challenge [53] in Table 9. The results from HEAR challenge reported the maximum of both settings ($L_1$ or $L_2$, $L_3$). Apart from Wav2vec2 which is trained on speech data, all other models were trained on non-speech audio. Pengi's linear probe $L_1$ and $L_3$ performance is consistently better than CLAP [15]. In the Sound Events and Music domain, Pengi outperformed other models. In the Speech Emotion domain, Pengi performed better than non-speech models but lower than models trained on speech (Wav2vec2). The experiment indicates that the next token prediction *does help* in learning audio representations useful for various domains.

| Model | Sound Events ↑ | | Music ↑ | | Speech Emotion ↑ | |
|---|---|---|---|---|---|---|
| | ESC50 | FSD50k | GTZAN Genres | Opera | RAVDESS | CREMA-D |
| YAMNet | 0.8375 | - | 0.847 | 0.9405 | 0.479 | 0.4533 |
| Open L3 | 0.7505 | 0.4470 | 0.879 | 0.9746 | 0.604 | 0.5497 |
| Wav2CLIP | 0.7589 | 0.3617 | 0.748 | 0.9363 | **0.684** | 0.5116 |
| PaNN | 0.9085 | - | 0.860 | 0.9112 | 0.429 | 0.5550 |
| Wav2Vec2 | 0.5610 | 0.3417 | 0.780 | 0.9067 | - | **0.6562** |
| CLAP ($L_1$) | 0.8995 | 0.5024 | 0.73 | 0.6399 | 0.4044 | 0.2315 |
| CLAP ($L_3$) | 0.9310 | 0.5690 | 0.8330 | 0.8263 | 0.4512 | 0.2830 |
| Pengi (ZS) | 0.9195 | 0.4676 | 0.3525 | 0.6229 | 0.2032 | 0.1846 |
| Pengi ($L_1$) | 0.8915 | 0.5608 | 0.8000 | 0.9193 | 0.4774 | 0.5057 |
| Pengi ($L_3$) | **0.9485** | **0.6235** | **0.9010** | **0.9883** | 0.6108 | 0.5916 |

Table 9: Shallow learning experiment where the audio encoder is frozen in all the experiments. ZS is zero-shot and $L_i$ indicates $i$ linear layers used. Unless specified, each model reports the best of $L_1$, $L_2$, and $L_3$.

## 6 Limitations

**Trade-off between close-ended and open-ended tasks performance.** The classification and text generation performance of Pengi is competitive against contrastive models. However, text-based retrieval performance lags behind that of contrastive models [11, 58]. Although these models excel at retrieval, they are limited to close-ended tasks. Thus, there is a trade-off between both types of learning methods proposed so far in the literature.

**Limitations inherent to Language Models.** Pengi benefits from the encyclopedic knowledge of pre-trained Language Models (LM). However, as pretrained LM is a component of Pengi, they also inherit their limitations. For example, LM are known to hallucinate [28] and specific to Pengi, can produce responses not grounded or conditioned on audio. Similarly, Pengi falls back to LM behavior if no audio is provided or if the audio knowledge is limited. Therefore, the risks of LM, namely propagating stereotypes, and biases and potentially producing offensive language are still applicable to Pengi. The recent works [48, 56] in the NLP field try to address these issues. However, specifically studying risks and limitations can uncover new insights that can accelerate the development of ALMs.

## 7 Conclusions

We proposed Pengi, a novel Audio Language Model that leverages Transfer Learning by framing all audio tasks as text-generation tasks. It takes as input, an audio recording, and a text prompt, and generates free-form text as output. Pengi is capable of handling both, close-ended and open-ended audio tasks. We benchmarked Pengi on 21 downstream tasks and show it yields SoTA performance in several of them. Our findings break ground in prompting language models with audio for general-purpose audio understanding.

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

Figure 4: More examples of audio and text prompt input and their corresponding textual responses. Images are for illustration purposes only.

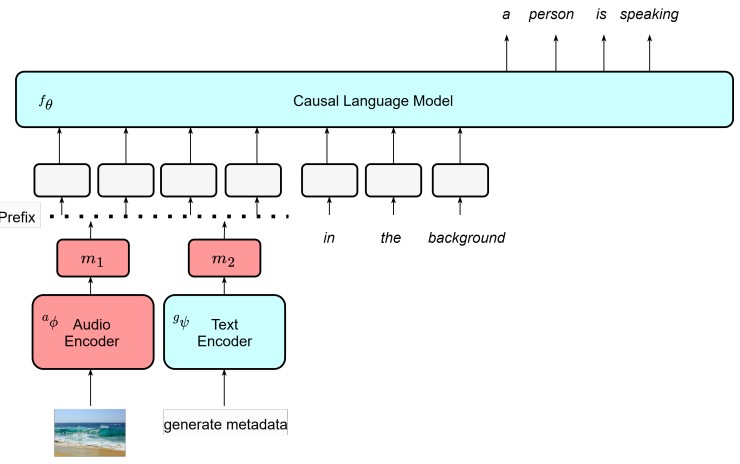

Figure 5: The user can also add an additional second text input and guide the output of Pengi. For example, the user can add "in the background" after the audio and text prefix and Pengi produces the output "a person is speaking". Compared to Fig 2, the output of Pengi changes to what the user has prompted in the second text input which is about background sounds.

# A    Additional text input

Pengi takes as input, an audio recording and text, and generates free-form text as output. During inference, an audio encoder $a_\phi$ and a mapping network $m_1$ represent each audio recording as a sequence of continuous embeddings. Similarly, a text encoder $g_\phi$ and a mapping network $m_2$ does the same for the corresponding text input. Both sequences are combined as a prefix to prompt a pre-trained frozen language model $f_\theta$. The language model generates tokens starting from the prefix.

The text input acts as task induction and helps guide the language model to produce the desired output. Let's take an example of human speech recording. A text input of "generate audio caption" will generate a caption like "a person speaking with a car moving in the background", while a text input of "this sentiment is" will produce a response like "negative". However, there are instances where we want to guide the language model further to answer or complete a specific query we had. We can do this by additional text input. This is depicted in Fig 5. The second text input gets tokenized by the frozen language model's tokenizer and converted into continuous embedding by the frozen language model's embedding function. Therefore, the new prefix consists of a sequence of embeddings associated with audio, first text input, and second text input which originates from the audio encoder, text encoder, and frozen language model's embedding function respectively.

Some examples and the effects of the second text input are shown in Fig 6. Empirically, we have seen the additional second input produces meaningful output only when used with text input of "generate metadata". The examples shown in Fig 6 are cherry-picked. The additional text input often causes Pengi to lose track of the audio data and hallucinate its own text or fall back to frozen language model behavior. It is not clear how to ground the output in audio information when additional text input is provided. Further investigation in this direction will enable new scenarios including in-context learning.

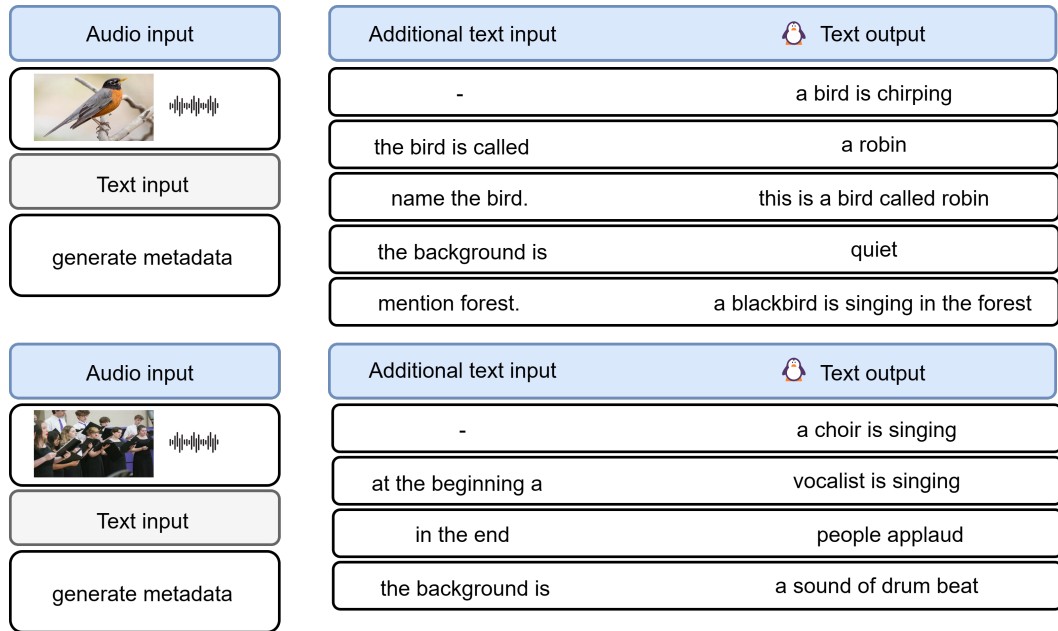

Figure 6: Examples of audio-text input with additional text input and their corresponding textual responses. Images are for illustration purposes only. The '-' symbol indicates additional text input was not used.

# B    Inferring audio prefix

The audio encoder and text encoder followed by mapping networks, jointly forms the prefix which prompts the frozen language model. To understand more about Pengi's natural language response, we try to interpret prefixes as a sequence of tokens or words. Each prefix embedding is mapped to the highest similarity token from the GPT2 vocabulary [42]. The similarity method used is cosine similarity. This is possible as the prefix and GPT2 embeddings occupy the same latent space. We

use this method on a few examples from the ESC50 dataset [45]. The examples of Pengi's generated output and the inferred audio prefix are shown in Table 10. The interpretations are hard to follow but do contain salient words that are related to audio content. For example, each inferred audio prefix contains words associated with content of audio like babies, thunder, chicken, etc which also appear in corresponding Pengi's natural language output.

One reason interpreted prefix does not have a clear structure is that the mapping network has to do two things at once - comprehend both the audio and text input and guide the fixed language model. Mokady et. al.[42] observed that the interpreted prefix is more comprehensible when GPT2 is also fine-tuned. A similar method can be followed to infer the text input prefix, but we didn't find any interpretable insights there.

| Text output | Inferred audio prefix |
|---|---|
| a **baby** is crying loudly and loudly | and, the my's the first the and and fixme the the supern the. coma the BST in in improvis the **babies** in in the noises from noises in the ( the the and innovative for |
| a **thunder** claps and then a **thunderstorm** hits | and- the bigHUD the the the and as"] the theth P the. weather the close andscape. **thunder** in- the Audiostorms interview click in the the and i unsettling, |
| a **rooster** is crowing loudly | and at the newone the new the and to OUR the theron the. **chickens** theities the in imperson the **chickens** to in the Audio sitcom. chickens in the ( the the, Mumbai the |
| a **bird** is **singing** in the background | and, **the great bird** the first the and and OUR the the number La the in **bird** the one great and photography and **bird** that. in Audio **owl** interview **singing** being: the and I innovative, |

Table 10: Examples of Pengi output and their corresponding inferred audio prefix. The input text prompt is "generate audio caption" for all examples. We bold the salient words relating to the input audio and text output.

## C  Effect of text prompts

The choice of input text prompt changes Pengi's downstream task performance. We analyze the performance of seven of the input text prompts defined in Section 4.1 for downstream tasks. For some tasks, only specific prompts are applicable, for example, *'question: {}'* prompt for AQA and *'this emotion is'* for emotion recognition. Pengi's performance on each downstream task corresponding to the different input text prompts is shown in Table 11). In summary, we see that the prompt *'generate metadata'* works well on average for close-ended downstream tasks.

| Downstream Dataset | Text prompts ↑ | | | | | | |
|---|---|---|---|---|---|---|---|
| | question: {} | generate audio caption | generate metadata | this is a sound of | this acoustic scene is | this music note is | this emotion is |
| Clotho Cap. | - | 0.2709 | - | - | - | - | - |
| AudioCaps Cap. | - | 0.4667 | - | - | - | - | - |
| ClothoAQA | 0.6453 | - | - | - | - | - | - |
| ESC50 | - | 0.8870 | 0.9195 | 0.6910 | - | - | - |
| FSD50k | - | 0.4676 | 0.4504 | 0.4572 | - | - | - |
| US8k | - | 0.7185 | 0.6585 | 0.5731 | - | - | - |
| DCASE17 | - | 0.3150 | 0.3143 | 0.3506 | - | - | - |
| AudioSet | - | 0.1216 | 0.1230 | 0.1635 | - | - | - |
| TUT 2017 | - | 0.2562 | 0.3525 | 0.2216 | 0.1716 | - | - |
| GTZAN Genres | - | 0.3230 | 0.3420 | 0.3180 | - | - | - |
| GTZAN MS | - | 0.9440 | 0.9606 | 0.9922 | - | - | - |
| Opera | - | 0.2373 | 0.6229 | 0.4449 | - | - | - |
| NSynth Instrument | - | - | - | - | - | 0.5007 | - |
| NSynth Pitch | - | - | - | - | - | 0.8676 | - |
| NSynth Velocity | - | - | - | - | - | 0.3728 | - |
| NSynth Qualities | - | - | - | - | - | 0.3860 | - |
| RAVDESS | - | - | - | - | - | - | 0.1846 |
| CREMAD | - | - | - | - | - | - | 0.2032 |
| Vocal Sounds | - | 0.5778 | 0.6035 | 0.5688 | - | - | - |
| SESA | - | 0.5162 | 0.5402 | 0.5350 | - | - | - |
| ESC50 Actions | - | 0.5277 | 0.5111 | 0.4846 | - | - | - |
| Clotho Ret. (T2A) | - | 0.0938 | - | - | - | - | - |
| AudioCaps Ret. (T2A) | - | 0.1771 | 0.1407 | - | - | - | - |
| Clotho Ret. (A2T) | - | 0.1148 | - | - | - | - | - |
| AudioCaps Ret. (A2T) | - | 0.1819 | 0.1771 | - | - | - | - |

Table 11: We use different text prompts and observe the performance on downstream tasks. '-' indicates the prompt is not used. The metrics used for each downstream tasks are same as Table 3.

# D Constrastive Learning model details

We follow and train a CLAP [15] model for the choice of contrastive model used in our experiments. We use transformer-based audio and text encoder. The audio encoder is HTSAT [6] and the text encoder is from CLIP [47]. Both the encoders are followed by a linear transformation called the projection layer. We finetune both the encoder and their projection layers. After contrastive training, the audio encoder and text encoder are used in Pengi.

Consider a batch size of $N$. Let the audio and text embedding be represented by $E_t \in \mathcal{R}^{N \times d}$ and $E_a \in \mathcal{R}^{N \times d}$. Then the resulting similarity matrix $C$ is:

$$C = \tau(E_t \cdot E_a{}^T) \tag{4}$$

We use the loss function ($\mathcal{L}$) of symmetric cross-entropy: projections

$$\mathcal{L} = 0.5(\ell_{text}(C) + \ell_{audio}(C)) \tag{5}$$

where $\ell_k = \frac{1}{N} \sum_{i=0}^{N} \log diag(softmax(C))$ along text and audio axis respectively.

**implementation details.** The audio is sampled at 44.1 kHz and is converted to a log Mel spectrogram with 64 Mel bins, a hop size of 320 secs, and a window size of 1024 secs in the range of 50-8000 Hz. We randomly truncate all audio files to 7 seconds in length for HTSAT. All models are trained with Adam Optimiser [32] for 45 epochs with a batch size of 1536 on 20 V100 GPUs. We use a linear schedule with 2000 warmup steps and a base learning rate of 1e-4.

**Results.** To verify the training, we check our CLAP's performance on the ESC50 dataset. The results are shown in Table 15.

| Model | ESC50 |
|---|---|
| Wav2CLIP | 0.414 |
| AudioCLIP | 0.694 |
| CLAP | 0.826 |
| LAION | 0.91 |
| CLAP (ours) | 0.89 |

Table 12: CLAP zero-shot performance on ESC50

# E Frozen audio encoder

The audio encoder $a_\phi$ transforms the raw audio input into an audio embedding. We used the audio transformer backbone from CLAP trained in Section D as our audio encoder in our experiments. In Computer Vision, Visual Language Models [42, 2, 39] use an image encoder from CLIP [47] which is frozen throughout experiments. However, there is a magnitude order difference in data collection of image-text vs audio-text pairs. Therefore, for Pengi we train the audio encoder as well. Nonetheless, we report numbers on Pengi's performance if the audio encoder is kept frozen. The results are shown in Table 13. Frozen Pengi underperforms Pengi across all downstream tasks.

| Model | Audio Captioning ↑ | | Audio Q&A ↑ | Sound Event Classification ↑ | | | |
|---|---|---|---|---|---|---|---|
| | AudioCaps | Clotho | ClothoAQA | ESC50 | FSD50K | US8K | DCASE17 Task 4 |
| Frozen Pengi | 0.4535 | 0.2577 | 0.6395 | 0.8950 | 0.4117 | 0.6319 | 0.3225 |
| Pengi | **0.4667** | **0.2709** | **0.6453** | **0.9195** | **0.4676** | **0.7185** | **0.338** |

| Model | Acoustic Scene Classification↑ | Music ↑ | | Instrument Classification ↑ | | Music Note Analysis↑ | | |
|---|---|---|---|---|---|---|---|---|
| | TUT2017 | Music Speech | Music Genres | Beijing Opera | Instrument family | NS. Pitch | NS. Velocity | NS. Qualities |
| Frozen Pengi | 0.3449 | 0.9219 | 0.2550 | 0.4814 | 0.2949 | 0.7131 | 0.3330 | 0.3830 |
| Pengi | **0.3525** | **0.9688** | **0.3525** | **0.6229** | **0.5007** | **0.8676** | **0.3728** | **0.3860** |

| Model | Emotion Recognition↑ | | Vocal Sound Classification↑ | Action Recog.↑ | Survei llance.↑ |
|---|---|---|---|---|---|
| | CRE MA-D | RAV DESS | Vocal Sound | ESC50 Actions | SESA |
| Frozen Pengi | 0.1816 | 0.1312 | 0.5371 | 0.5196 | 0.5316 |
| Pengi | **0.1846** | **0.2032** | **0.6035** | **0.5277** | **0.5402** |

Table 13: The model 'Frozen Pengi' indicates Pengi with audio encoder frozen. The '-' symbol indicates numbers were not available while '✗' indicates that the model cannot support the task. Higher is better for all numbers. The evaluation metric is mAP for FSD50k, AudioSet, and NSynth sonic; F1 score for DCASE17; and SPIDEr for AudioCaps and Clotho captioning. All other downstream tasks use Accuracy.

| Model | Audio Captioning ↑ | | Audio Q&A ↑ | Sound Event Classification ↑ | | | |
|---|---|---|---|---|---|---|---|
| | AudioCaps | Clotho | ClothoAQA | ESC50 | FSD50K | US8K | DCASE17 Task 4 |
| Exp B | **0.4857** | 0.2545 | 0.6316 | 0.9215 | 0.4478 | 6882 | 0.3314 |
| Pengi | 0.4667 | **0.2709** | **0.6453** | **0.9195** | **0.4676** | **0.7185** | **0.3380** |

| Model | Acoustic Scene Classification↑ | Music ↑ | | Instrument Classification ↑ | | Music Note Analysis↑ | | |
|---|---|---|---|---|---|---|---|---|
| | TUT2017 | Music Speech | Music Genres | Beijing Opera | Instrument family | NS. Pitch | NS. Velocity | NS. Qualities |
| Exp B | 0.3241 | **0.9609** | 0.317 | 0.6864 | 0.5 | 0.8591 | 0.3708 | 0.377 |
| Pengi | **0.3525** | 0.9688 | **0.3525** | **0.6229** | **0.5007** | **0.8676** | **0.3728** | **0.386** |

| Model | Emotion Recognition↑ | | Vocal Sound Classification↑ | Action Recog.↑ | Surveillance.↑ |
|---|---|---|---|---|---|
| | CREMA-D | RAVDESS | Vocal Sound | ESC50 Actions | SESA |
| Exp B | 0.1728 | 0.1769 | 0.5798 | 0.5282 | 0.4923 |
| Pengi | **0.1846** | **0.2032** | **0.6035** | **0.5277** | **0.5402** |

Table 14: Exp B is Pengi with mapper $m_2$ but without the text encoder. The evaluation metric is mAP for FSD50k, AudioSet, ESC50-Actions, and NSynth sonic; F1 score for DCASE17; and SPIDEr for AudioCaps and Clotho captioning. All other downstream tasks use Accuracy.

# F    Effect of text encoder

Pengi's architecture in Figure 2 consists of a text encoder $g_\psi$ that transforms the input text into text embeddings. Then a mapping network $m_2$ converts these embeddings into a sequence of k embeddings. A natural question that arises here is *"Why is an explicit mapping needed for input text?"*. We conducted two experiments to evaluate the effect of omitting $m_2$ and/or the text encoder. We denote Exp A as Pengi without the text encoder and $m_2$ (input text directly to LM), and Exp B as Pengi without the text encoder but with $m_2$ (input text to $m_2$). In Exp A, we found that removing resulted in a loss of coherence between the input text prompt and the output text. For example, an input prompt about identifying an emotion class "the emotion is " resulted in random text output and thus random performance. In Exp B, we removed the text encoder but retained $m_2$. The Exp B architecture is depicted in Fig 7 and its results are shown in Table 14. By removing the text encoder, the model performs slightly lower than the proposed architecture with both components.

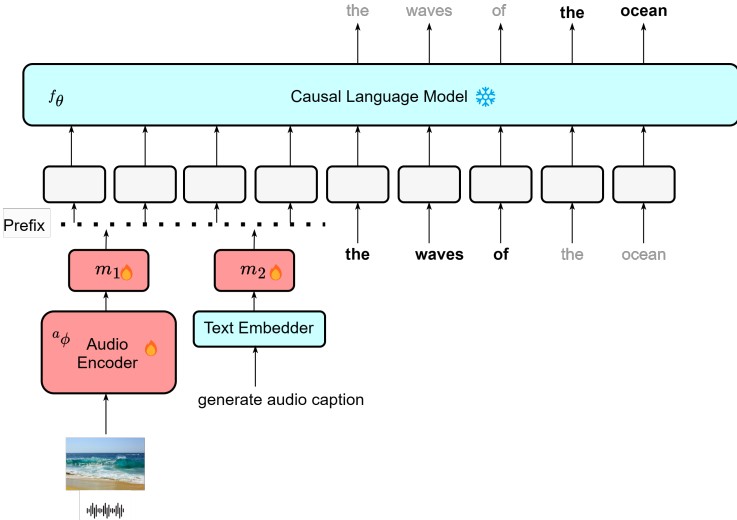

Figure 7: Pengi architecture without the text encoder $g_\psi$. The text prompt is tokenized and embedded by text embedder, followed by the mapping network $m_2$. The results of this architecture are shown in Table 13

| | Audio Captioning ↑ | | Audio Q&A ↑ | Sound Event Classification ↑ | | | |
|---|---|---|---|---|---|---|---|
| Model | AudioCaps | Clotho | ClothoAQA | ESC50 | FSD50K | US8K | DCASE17 Task 4 |
| CLAP* | ✗ | ✗ | ✗ | 0.8916 | 0.3398 | **0.7661** | **0.3387** |
| Pengi | **0.4667** | **0.2709** | **0.6453** | **0.9195** | **0.4676** | 0.7185 | 0.3380 |

| | Acoustic Scene Classification↑ | Music ↑ | | Instrument Classification ↑ | | Music Note Analysis↑ | | |
|---|---|---|---|---|---|---|---|---|
| Model | TUT2017 | Music Speech | Music Genres | Beijing Opera | Instrument family | NS. Pitch | NS. Velocity | NS. Qualities |
| CLAP* | 0.3037 | **1.0** | **0.479** | 0.4025 | 0.415 | 0.1337 | 0.2185 | 0.2545 |
| Pengi | **0.3525** | 0.9688 | 0.3525 | **0.6229** | **0.5007** | **0.8676** | **0.3728** | **0.386** |

| | Emotion Recognition↑ | | Vocal Sound Classification↑ | Action Recog.↑ | Surveillance.↑ |
|---|---|---|---|---|---|
| Model | CREMA-D | RAVDESS | Vocal Sound | ESC50 Actions | SESA |
| CLAP* | 0.1512 | 0.1692 | 0.5522 | 0.508 | **0.7094** |
| Pengi | **0.1846** | **0.2032** | **0.6035** | **0.5277** | 0.5402 |

Table 15: We train a new CLAP* model on the same 3.4M pairs training data used Pengi. The '✗' indicates that the model cannot support the task. Higher is better for all numbers. The evaluation metric is mAP for FSD50k, AudioSet, ESC50-Actions, and NSynth sonic; F1 score for DCASE17; and SPIDEr for AudioCaps and Clotho captioning. All other downstream tasks use Accuracy.

## G   Different type of Pengi errors

There are three types of errors that lead to a drop in Pengi's performance. We categorize them into audio concept errors, hierarchy errors, and text-matching errors.

**Audio concept errors.** These types of errors are when the model gets the base audio concepts wrong. For example, while generating an audio caption, the model predicts it as "a sound of a dog barking in a neighboring field" instead of "a sound of door knocks with cars moving nearby". This indicates the model fails to detect the sound event of a door knock and confuses it with dog barking. These are Pengi model errors stemming from the audio encoder.

**Heirarchy errors.** The hierarchy error comes from a mismatch between Pengi's model prediction and the target domain classification. For example, in classifying sound events, Pengi predicts the sound as "domestic sounds", however for ESC50, the target classification requires a more fine-grained classification within domestic sounds like Vaccum cleaner, Toilet flush, brushing teeth, etc. If text matching is used for classification, then the model will not be able to categorize "domestic sounds" into any of the fine-grained classes. To solve this error and get a more fine-grained response, we can use improved text prompts or switch to the log-likelihood method.

**Text-matching errors.** The text-matching errors are the errors that result from the text embeddings or the text-matching method used. This means depending on the text embedding and similarity method used, the performance of Pengi on close-ended tasks will change.

## H   Constrastive Learning and Generative Pretraining

We compare our model Pengi with CLAP [15], a state-of-the-art Zero-Shot model that has been evaluated on 16 downstream tasks. However, CLAP is trained on a smaller amount of audio-text data. This leads us to ask: *"Is the improved performance due to the larger training data or the generative pretraining?"*. We already know that generative pretraining allows us to perform open-ended tasks like Audio Captioning, AQA, which are not possible with contrastive models. But this does not tell us if: *generative pretraining is beneficial for close-ended tasks like classification?*. To answer this question, we train a CLAP model with the same data 4.1) that we use to train Pengi. We call this model CLAP*.

**Results.** The results are shown in Table 15. We see generative pertaining (Pengi) outperforming contrastive learning (CLAP*) on average. Moreover, with generative pretraining, the model can perform open-ended tasks like Audio Captioning and Audio Question Answering.

An interesting observation is Pengi outperforms human performance (81%) on ESC50. Humans have limitations inherent to how much information a participant can handle at once. In the case of ESC50, humans listen to the audio once, and have to remember the audio content, task description, and choose among 50 different classes. Moreover, listeners have different degrees of familiarity with prototypical content from different sound classes, whereas Pengi has been exposed to similar content during training. In a sense, Pengi is an expert listener, whereas the humans in the listening experiment were not.

