# OpenReview forum: "Pengi: An Audio Language Model for Audio Tasks"
_NeurIPS.cc/2023/Conference — NeurIPS 2023 poster_

### Official Review · Reviewer_4yQP · 2023-07-06

**Soundness:** 4 excellent
**Presentation:** 2 fair
**Contribution:** 4 excellent
**Rating:** 5
**Confidence:** 4

**Summary:**

Pengi is a novel audio language model that leverages transfer learning to frame all audio tasks as text-generation tasks. It takes an audio recording and text as input, and generates free-form text as output. The input audio is represented as a sequence of continuous embeddings by an audio encoder, and the text input is represented similarly by a text encoder. Both sequences are then combined as a prefix to prompt a pre-trained frozen language model. When evaluated on 22 downstream tasks, Pengi achieved state-of-the-art performance in several of them.

**Strengths:**

1. The proposed model is a novel audio language model that can be used for multiple audio tasks, including close-ended and open-ended tasks. It does not require any additional fine-tuning or task-specific extensions.

2. The paper introduce a new learning framework that frames all audio tasks as audio and text input to text output tasks. This framework uses a single training procedure and a captioning objective function. For training, Pengi uses new audio task templates inspired by Instruction Tuning.

3. Pengi has been extensively evaluated on 21 (or 22?) downstream tasks across various audio domains. It achieved state-of-the-art performance in several of these tasks, establishing a baseline for general-purpose ALM.

**Weaknesses:**

.

**Questions:**

1. Please add more baseline - a cascade model for each task.
2. Does the model has capabilities of ASR?
3. Can you add ablation analysis of each component?

**Limitations:**

.

---

> ### Author Rebuttal · Authors · 2023-08-08
>
> We sincerely thank the reviewer for recognizing our contribution! We address every question and hope that our response resolves your concerns. Any follow-up questions are welcome.
> ***
> **Questions 1. Please add more baseline - a cascade model for each task.**\
> For Table 3, we chose CLAP because it is the only Zero-Shot model with a comprehensive evaluation (16 downstream tasks). The next best evaluation was only on 8. Thus, providing no evidence of performance across other domains like speech and music, which tend to be the most difficult. For the rest of the Tables, we compared against SoTA results even if it came from different models and learning methods. We compared against SoTa Zero-Shot models in Table 8, a subset of Table 3, for Sound Event Classification. Even against SoTa from supervised learning models in Table 5 and 7 for Audio Q&A and Audio Captioning respectively. Table 9, against SSL, supervised and trained on speech audio. Training with ensemble models (cascade) will provide insights into how methods compliment, but it was our scope.
>
> **Questions 2. Does the model has capabilities of ASR?**\
> Pengi was not trained on any speech audio-transcript data, so it does not support ASR. We believe that the key to integrating audio and speech tasks is to develop a universal audio encoder architecture, which is an exciting and important direction for the future, but beyond the scope of this work.
>
> **Questions 3. Can you add ablation analysis of each component?**\
> We performed different ablation studies and a new experiment for text encoder:
> 1) We study the choice of text encoder and its mapper- m2 for Pengi. We denote exp A as Pengi without both the text encoder and m2 and exp B as Pengi without the text encoder. In exp A, we found that removing both m2​ and the text encoder resulted in a loss of coherence between the input text prompt and the output text. For example, an input prompt about identifying an emotion class "the emotion is " resulted in random text output and thus random performance. In exp B, we found comparable performance to Pengi. We attach this as Table 1 in PDF in the global response.
> 2) We analyzed the choice of text prompt in performance in Appendix Section C, and found that the prompt "generate metadata" is a good default choice.
> 3) We analyzed the prefix output from the audio encoder-mapper in Appendix Section B and found that it contains relevant keywords present in the output text, but can be noisy.
> 4) We analyzed freezing and unfreezing the audio encoder in Appendix Section E, and found that unfreezing the encoder yielded better performance.

---

### Official Review · Reviewer_sUJx · 2023-07-06

**Soundness:** 2 fair
**Presentation:** 4 excellent
**Contribution:** 2 fair
**Rating:** 5
**Confidence:** 5

**Summary:**

This paper explores Large Language Models (LLMs) in the context of audio processing. The core idea is to do audio-injected instruction tuning for pre-trained LLM. The method is simple: collecting a lot of audio-text paired data and use it to fine-tune a pre-trained CLAP audio encoder together with a frozen LLM. Audio is injected by concatenating the CLAP feature sequence before text. Taking advantage on the nature of LLMs training strategy, the pair relation can be provided in any form (e.g., captioning, label, metadata) and simply put into a pre-defined template to train with next token prediction. Evaluation is done on a wide collection of audio-related tasks using the output sentence of the LLM given the input audio and question.

**Strengths:**

- This is the first (together with few concurrent papers) work to explore LLM for audio processing.
- Performance on audio captioning is solid.
- The method is simple in a good way, and this work can be served as a baseline for audio processing with LLMs if the evaluation can be more completed (see weaknesses).

**Weaknesses:**

The biggest concern I have for this paper is that **the evaluation seems to be limited/biased**.

- Table 3: In Pengi's framework, CLAP serves as a feature extractor that is pre-trained and unfrozen. It is obvious that Pengi can (and should) be making improvement over CLAP considering the scale of the system, the total amount of data used, and the computational power required.  Comparing Pengi to state-of-the-art methods on each bench mark would better justify its value.  (Imho, this is also why LLM is a great success - they generalized to and performed well on most of the tasks so good to the point where people can live with the amount of data/computation it costs.) Even if the numbers are not overwhelming, it will still add value to this work as a first step of exploring LLM's application in audio.
- Table6/Section5.4: It is unclear how the authors obtained the retrieval performance for existing contrastive methods using their text-to-text retrieval pipeline. How do they index the dataset with text given the audio/text encoder from contrastive learning? Moreover, the text-to-text setup could be biased and unfair for other methods, since Pengi builds on top of pre-trained LLM and focuses on text during training. While a 100% fair comparison might not be possible, it is clearly unfair to use significantly more resource AND compare in a way favoring the large model. Table 5/7 are good examples where at least the evaluation protocol is fair and I don't see a reason why Table 6 should be treated differently. Again, even if the numbers are not good enough, it would still help the community by establishing a standard for general purpose audio model.

Overall, this work is interesting in terms of exploring general audio processing model, but the evaluation should be improved. I would happily raise my score if the above-mentioned concerns can be addressed.


**Questions:**

- The use of text encoder: line 112 said it's fundamental but there seems to be no ablation on this. Having LLMs taking text as input seems to be more intuitive. Any idea why?
- For line 270: "... by 32%", isn't this 22-ish%?


**Limitations:**

- Generalizability (data): Table 8 shows that in the cases (namely, ESC50, US8K, and DCASE17) where Pengi have not seen the dataset, results are not as competitive. It would be good if more details on evaluation pipeline can be provided and interesting to see some error analysis.
- Generalizability (format): All the tasks this paper considered are covered by the templates hence the input is not in free form. One of the key strength of instruction-tuned LLM is the ability to take any input and answer accordingly. are there examples where the question (text input) is never seen during training?

---

> ### Author Rebuttal · Authors · 2023-08-08
>
> We sincerely thank the reviewer for recognizing our contribution and novelty! We take every comment seriously and address every concern point by point. We hope that our response resolves your concerns. Any follow-up questions are welcome.
> ***
> **Questions 1. Table 3: In Pengi's framework, CLAP serves as a feature ... it will still add value to this work as a first step of exploring LLM's application in audio.**\
> As the reviewer suggested, our goal is to add value exploring LLM's for audio rather than establishing SoTa in every task. For Table 3, we chose CLAP because it is the only Zero-Shot model with a comprehensive evaluation (16 downstream tasks). The next best evaluation was only on 8. Thus, providing no evidence of performance across other domains like speech and music, which tend to be the most difficult. For the rest of the Tables, we compared against SoTA results even if it came from different models and learning methods. We compared against SoTa Zero-Shot models in Table 8, a subset of Table 3, for Sound Event Classification. Even against SoTa from supervised learning models in Table 5 and 7 for Audio Q&A and Audio Captioning respectively. Table 9, against SSL, supervised, and trained on speech audio. To enhance our baseline from Table 3, we trained a CLAP model on the same amount of pairs data as Pengi (3.4M) to put aside variations due to data. Pengi's strong performance still holds. We attach this as Table 2 in PDF in the global response.
>
> **Questions 2. Table6/Section5.4: It is unclear how the authors obtained the retrieval performance ... it would still help the community by establishing a standard for general purpose audio model.**\
> We acknowledge the reviewer’s point. Models trained with contrastive learning learn the similarity between audio and text and can do audio-to-text and text-to-audio retrieval in one step. Text generation models like standard audio captioning models and Pengi, don't learn the multimodal similarity and thus require additional steps to compare text and audio. Therefore, in Table 6, we compared Pengi using a setup proposed by Kim et al. [1] and the best-performing models using the same setup. Due to the inherent difference between contrastive and generative models, there is no fair comparison. However, for a purely numerical comparison, contrastive learning methods surpass all text-generation methods (Pengi or others) for the retrieval task. We explain this in Section 6 paragraph 1. We will include these numbers in Table 6 to provide a reference to the readers.
>
> [1] Kim, Minkyu, Kim Sung-Bin, and Tae-Hyun Oh. "Prefix tuning for automated audio captioning." ICASSP 2023-2023 IEEE International Conference on Acoustics, Speech and Signal Processing (ICASSP). IEEE, 2023.
>
> **Questions 3. The use of text encoder: line 112 said it's fundamental but there seems to be no ablation on this. Having LLMs taking text as input seems to be more intuitive. Any idea why?**\
> We appreciate the reviewer's question. We conduct an experiment and discuss the findings and results in global response question 1.
>
> **Questions 4. For line 270: "... by 32%", isn't this 22-ish%?**\
> We are sorry for any misunderstanding. We report relative percentage improvements and not absolute percentage improvements. Since AudioCLIP and Pengi score 0.694 and 0.92 respectively on ESC50, we calculate the relative improvement as (0.92-0.694)/0.694 = 32.5%.
>
> **Questions 5. Generalizability (data): Table 8 shows that in the cases (namely, ESC50, US8K, and DCASE17) where Pengi have not seen the dataset, results are not as competitive. It would be good if more details on evaluation pipeline can be provided and interesting to see some error analysis.**\
> We evaluated the generalizability (data) of Pengi by testing it on 22 downstream tasks (Table 3) and comparing its performance with different SoTA models in the literature (Table 4-9). Pengi’s zero-shot performance achieves SOTA on several but not all downstream tasks. Table 8, a subset of Table 3, evaluates zero-shot sound event classification on 4 tasks and includes the best models in the literature. None of the models, including Pengi, have seen the dataset during training. We believe Pengi's results are competitive at 92% vs SoTA of 91% for ESC50 and 72% vs SoTA of 77%. To better understand the errors made by Pengi, we conduct an error analysis. We identify three types of errors that cause a decline in Pengi’s performance. We classify them into audio concept errors, hierarchy errors, and text-matching errors and describe them in Appendix Section F.
>
> **Questions 6. Generalizability (format): All the tasks this paper considered are covered by the templates hence the input is not in free form. One of the key strength of instruction-tuned LLM is the ability to take any input and answer accordingly. are there examples where the question (text input) is never seen during training?**\
>  We apologize for any confusion and we would clarify in the manuscript that:
>  - Pengi can handle different ways of phrasing the same prompt, thanks to its text encoder. For instance, it can recognize that “this is a sound of” and “detect sound events” are asking for the same kind of output (e.g.“dog barking") even if the template only has “this is a sound of”. If the prompt is completely different from the ones it was trained on, Pengi defaults to generating metadata as a general response.
> - We agree with the reviewer that it is an interesting challenge to make LLMs respond to any input appropriately. We have explored this direction in Appendix Section A, where we show that the user can use a default text prompt like “generate metadata” and then provide more information or ask follow-up questions. This enables the user to steer the conversation with additional unseen prompts (Fig. 6), such as “the background is”, “mention forest.”, etc. We acknowledge that this is not Pengi's strength. The literature suggests that scaling up training data could improve this issue.

---

> > ### Comment · Reviewer_sUJx · 2023-08-17
> >
> > I would like to thank the authors for answering the questions. Overall, I am satisfied with the extra detail provided that complements this work. I will increase my rating to borderline accept.

---

### Official Review · Reviewer_vWxh · 2023-07-09

**Soundness:** 4 excellent
**Presentation:** 4 excellent
**Contribution:** 3 good
**Rating:** 7
**Confidence:** 5

**Summary:**

This work is inspired by the visual language models (VLM) in the literature and presents an audio language model (ALM) for various audio tasks, including both open-ended and close-ended tasks. It also presents a comprehensive evaluation of the proposed ALM on a range of both open-ended and close-ended tasks, and showed very promising results.



**Strengths:**

This paper presents an innovative audio language model which leverage a pretrained audio encoder and a pretrained language model. Though the proposed network architecture is largely borrowed from vision language models like [47], it is still relatively new to the audio domain.

This paper is well-written and presents comprehensive evaluations on 22 downstream tasks.

**Weaknesses:**

I feel this work is somewhat limited in that it only uses a relatively small language model (GPT-2 line, 124M params) and there is no study of how the language model can affect the performance on the downstream tasks. It is possible that with a stronger language model, the performance of the proposed model on open-ended tasks can be further improved, while the performance on close-ended tasks may more depending on the audio encoder quality.

**Questions:**

- I feel it is a little bit counter-intuitive that in Figure 2, a `m2` mapping network is used after the text encoder, as text encoder  in Figure 2 is already encoded the text prompt into the same space as the text in the response part. What's the purpose of  `m2` here and do you have experimental results to show using `m2` is helpful ?

- It is also unclear to me how the network handle the variable length in text prompt. It looks like to me that both audio prompts and  text prompts in this works has a fixed length, i.e., 40 tokens for each part. However, it is unclear from the paper, how the authors would handle variable lengths of text prompt. It is also unclear what the effect of this fixed length (40 in the experiments) on the audio task performance.

- The authors have proposed 2 methods for evaluating the proposed model's performance on close-ended tasks: log-likelihood and text matching. However, all the following experiments using text matching and there is no comparison of log-likelihood based method vs text matching methods. Are they give relatively close results?


**Limitations:**

The authors have addressed the limitations of the proposed methods adequately.

---

> ### Author Rebuttal · Authors · 2023-08-08
>
> We sincerely thank the reviewer for recognizing our contribution and novelty! We hope that our response resolves your concerns. Any follow-up questions are welcome.
> ***
> **Questions 1. I feel this work is somewhat limited in that it only uses a relatively small language model (GPT-2 line, 124M params) and there is no study of how the language model can affect the performance on the downstream tasks. It is possible that with a stronger language model, the performance of the proposed model on open-ended tasks can be further improved, while the performance on close-ended tasks may more depending on the audio encoder quality.**\
> We appreciate the reviewer’s comment. Our default language model is GPT-2 base (124M), but we also tested GPT2-XL (1.5B). A larger LM improved performance on the open-ended task of Audio Q&A but have mixed results on Audio Captioning --one dataset improved the other one worsen. For close-ended tasks, there was no significant change. We attach the AQA results below. We agree the audio encoder does impact task performance, and presents an exciting direction for future work.
> | LLM | Parameters | Audio Q&A|
> ------|------------|----------|
> | GPT2-base | 128M |  0.645   |
> | GPT2-XL   | 1.5B |  0.701   |
>
> **Questions 2. I feel it is a little bit counter-intuitive that in Figure 2, a m2 mapping network is used after the text encoder, as text encoder in Figure 2 is already encoded the text prompt into the same space as the text in the response part. What's the purpose of m2 here and do you have experimental results to show using m2 is helpful ?**\
> We appreciate the reviewer's question. We conduct an experiment and discuss the findings and results in global response question 1.
>
> **Questions 3. It is also unclear to me how the network handle the variable length in text prompt. It looks like to me that both audio prompts and text prompts in this works has a fixed length, i.e., 40 tokens for each part. However, it is unclear from the paper, how the authors would handle variable lengths of text prompt. It is also unclear what the effect of this fixed length (40 in the experiments) on the audio task performance.**\
> The text encoder can handle input text of variable length. It produces sentence-level embedding for the input text. The mapping network $m_2$ then converts the sentence-level embedding into a fixed-length sequence of embeddings (prefix). The prefix length is a hyperparameter that we set to 40. Likewise, the model can deal with audio input of variable duration, and it maps the audio representation to a prefix of length 40. We empirically found that a prefix size of 40 achieved the best performance compared to a prefix size of 20 or 80.
>
> **Questions 4. The authors have proposed 2 methods for evaluating the proposed model's performance on close-ended tasks: log-likelihood and text matching. However, all the following experiments using text matching and there is no comparison of log-likelihood based method vs text matching methods. Are they give relatively close results?**\
> We chose the text-matching method for experiments because it is computationally less expensive. The log-likelihood method requires more computation and has lower performance on most datasets than the text-matching methods. However, the log-likelihood method has an advantage on out-of-domain or rare words that the text encoder cannot recognize. For instance, when identifying specific bird species by their song, the log-likelihood method outperforms the text-matching methods. We concur that a large-scale study is necessary to determine which evaluation method is superior.

---

> > ### Comment · Reviewer_vWxh · 2023-08-20
> >
> > Thanks for very much for the detailed explanation!

---

### Official Review · Reviewer_o28f · 2023-07-10

**Soundness:** 3 good
**Presentation:** 3 good
**Contribution:** 2 fair
**Rating:** 4
**Confidence:** 4

**Summary:**

This paper proposes a new audio-language learning model by treating existing audio tasks as text-generation tasks. The model architecture allows both open-ended and close-ended tasks. By evaluating on 22 downstream tasks, this paper shows competitive performance on many of them.

**Strengths:**

The idea of treating various forms of audio tasks as text generation task is reasonable, which allows scaling the size of the data to large models.

The evaluation is pretty extensive. Authors compared the proposed model on multiple benchmarks against multiple models.

The paper is well-written and easy to follow.

**Weaknesses:**

It seems that the strongest baseline is LAION CLAP [53], while authors did not compare with it in Table 3 or 9. In Table 8, Pengi did not outperform LAION, which seems to indicate that this model does not outperform the SOTA approach.

My other concern is that the model's performance is largely based on the LLM and thus limited to their weaknesses. This has been discussed in the limitation section as well.

**Questions:**

My main concern about this work is its performance against SOTA. I'd appreciate authors' clarification on that.

==== post rebuttal ===
I'd like to thank the authors for their responses. However, the performance improvement against LAION is not really convincing and comparisons are lacking, and thus I'm keeping the original score.

**Limitations:**

It has been discussed in Section 6.

---

> ### Author Rebuttal · Authors · 2023-08-08
>
> We thank the reviewer for recognizing our contribution and providing constructive feedback! We address every question and hope that our response resolves your concerns. Any follow-up questions are welcome.
> ***
> We present a novel and unified model that can handle both open-ended and close-ended audio tasks without relying on external modules or fine-tuning. Our key contributions are: (1) Pengi, the first Audio Language Model (ALM) in the literature, (2) audio task templates inspired by Instruction Tuning, and (3) a method to perform close-ended tasks with ALM. We evaluate Pengi on 22 downstream tasks and show that it achieves state-of-the-art results on most of them. Therefore, our main goal is not to beat SoTA all around with a single model, but to demonstrate the versatility and strength in performance of our method.
>
> **Questions 1. It seems that the strongest baseline is LAION CLAP [53], while authors did not compare with it in Table 3 or 9. In Table 8, Pengi did not outperform LAION, which seems to indicate that this model does not outperform the SOTA approach.**\
> We chose CLAP because it is the only Zero-Shot model with a comprehensive evaluation (16 downstream tasks). It is not conclusive if LAION will outperform Pengi in every task. First, the LAION model was only tested on 4 datasets related to sound event classification. Thus, providing no evidence of performance on multilabel classification (FSD50K) and across other domains like speech and music, which tend to be the most difficult. Second, in Table, 8 Pengi underperformed LAION on US8K but outperformed LAION on ESC50. LAION model cannot perform Audio Captioning or Audio Q&A without additional modules and finetuning.
>
> **Questions 2. My other concern is that the model's performance is largely based on the LLM and thus limited to their weaknesses. This has been discussed in the limitation section as well.**\
> We agree with the reviewer. All the limitations of LLMs inherent to LLMs apply to Pengi as well. We explore this in Section 6.
>
> **Questions 3. My main concern about this work is its performance against SOTA. I'd appreciate authors' clarification on that.**\
> In addition to our comment in Response 1, we compared against SoTa performance throughout our paper. We compared against SoTa Zero-Shot models in Table 8, a subset of Table 3, for Sound Event Classification. Against SoTa from supervised learning models in Table 5 and 7 for Audio Q&A and Audio Captioning respectively. Against SSL, supervised, and trained on speech audio models in Table 9.
>
> To enhance our baseline from Table 3, we trained a CLAP model (CLAP*) on the same amount of pairs of data as Pengi (3.4M) to put aside variations due to data. Pengi's strong performance still holds. We attach this as Table 2 in PDF in the global response.

---

### Official Review · Reviewer_iz2N · 2023-07-27

**Soundness:** 3 good
**Presentation:** 3 good
**Contribution:** 3 good
**Rating:** 6
**Confidence:** 4

**Summary:**

The authors present an approach to combine multiple non-ASR audio tasks into a single model. Motivated by audio language models and VLM, the authors propose using a frozen LLM to create an audio LM that can be used for open-ended (purely generative) and close-ended (classification) tasks. The main idea is to use an audio encoder and a text encoder to create fixed length prompts that can be used with a frozen LM after training. The authors train various tasks by using task specific text prompts (“generate audio caption”, “this is the sound of”, etc.). In this sense, the model is similar to a multitask learning model (more on this below). Results show that the model is competitive compared to other approaches in the literature that are task-specific, and an alternative multi-task model, CLAP.

**Strengths:**

- A single model that can handle multiple tasks, both open-ended and close-ended.
- One of the first approaches that incorporate VLM / LLMs to create a general purpose audio LM tasks.
- Results show gains using LLM for building the model, compared to a baseline that doesn't.

**Weaknesses:**

- While the model can cover a number of tasks with competitive results, the model still performs worse on some of the tasks considered compared to task-specific models.
- Furthermore, it is unclear if the method generalizes to new tasks, which is a main strength of non-audio LLMs / VLMs. The authors show zero-shot capabilities for new labels within an existing task. Although impressive, not entirely novel given prior works have approached the problem along similar veins (like CLAP, e.g.) but in limited settings. The results in Tab. 9 also show poor performance in zero-shot settings for a new task.
- The results in Tab. 8 also show that the model is not as best as some of the other models on zero-shot classification task.

**Questions:**

1) The authors tokenize and map text prompts. Why is an explicit mapping needed? Can’t they re-use the same tokenizer as the LLM?
2) Why does the audio and text prompt prefix have to be of fixed length? Does the length of the input audio affect quality?
3) Line 180 and Fig. 3: Text matching is not appropriately described and is a little hard to follow. Consider providing a more detailed description. Are the embeddings created for each token or for the entire sentence? If for the sentence, how are they created (summarized)?
4) The authors claim sota in a few different close-ended tasks. Are the gains coming from the new formulation (audio LM) or the extra datasets that the authors are now using? For example, what if the authors train a single model with multiple classification heads for the N-tasks (close-ended, at least)? Does it work worse than the current model that uses the frozen LLM and the proposed architecture?
5) Line 181: When computing log likelihoods, do the authors normalize based on the number of tokens in the desired values (dog barking vs. sea, e.g.)? Does the number of tokens affect the overall score?
6) Sec. 5.5: Does the presented result show that perhaps a good portion of the performance comes from training data. For instance, wav2vec2 works better on emotion recognition, likely because the training data is relevant. Are the training set the same for the remaining non-speech models in Tab. 9?

Minor
1) The authors use close-ended and open-ended tasks a lot, but only define it Sec. 4.2. Consider moving this to an earlier section to avoid confusion.
2) The authors say the text encoder is frozen (Line 111). Where does this encoder come from?
3) Why are the results in Tab. 4 for the best setting, worse than those in Tab. 7?
5) Please provide more descriptive captions for Tab. 4 – 6.
6) Line 271: Why does Pengi outperform human performance by such a large margin? It is interesting, so perhaps an explanation is warranted.
7) It would be useful to include the best results from Mei et al to Tab. 8. Also, it’d be useful to include some representative contrastive methods to Tab. 6 for audio retrieval.
8) Why does Pengi work better on some zero-shot tasks and not the others, compared to techniques like CLAP? Is it related to the training data?

Typos:
1) Line 51: For example, -> Examples include
2) Line 70: Missing citation (?)
3) Figure 2 caption: a text a prompt -> a text prompt
4) Line 151: question: question -> question: {question}
5) Line 205: 320 secs / 1024 secs: Perhaps the authors mean milliseconds?
6) Line 270: Missing Table# (Tab. 8?).

**Limitations:**

1) Does not support ASR, which is, arguably, one of the most important audio task.
2) Unclear if the model can learn new tasks. The existing tasks use a fixed prefix prompt, which is tokenized and mapped. This makes it cumbersome to add new tasks to the model. This is a significant drawback compared to current LLMs that can learn new tasks by prompting.
3) How much does the LLM help? What is the quality if the LM component is trained from scratch with the current dataset?

---

> ### Author Rebuttal · Authors · 2023-08-08
>
> We thank the reviewer for recognizing our contribution and providing detailed comments. We hope that our response resolves your concerns and welcome any follow-up questions.
> ***
> **Q1. The authors tokenize and map ... re-use the same tokenizer as the LLM**\
> We conduct an experiment and discuss the findings and results in global response question 1.
>
> **Q2: Why does the audio and text prompt ... of the input audio affect quality?**\
> We choose the audio and text prompt prefix to be of length 40. The length choice is a hyperparameter and does not have to be fixed. On the second question, yes, the length of the input audio affects downstream task performance. The model is trained with 7-second audio clips, so it performs best for audios of similar duration. Shorter audio degrades performance, especially for long sound events like bell gong.
>
> **Q3: Line 180 and Fig. 3: Text matching is not appropriately described ... they created (summarized)?**\
> We apologize and will revise the description. For the text matching, the text encoder generates a sentence-level embedding for Pengi's textual output. The sentence-level embedding corresponds to the embedding of the [CLS] token from the text encoder.
>
> **Q4: The authors claim sota in a few different ... the proposed architecture?**\
> We appreciate the reviewer’s question. More objectives and parameters may improve training, but only if the model can converge with multiple losses.  Also, adding heads is complex and hard to scale. For example, we have 22 tasks, so we need 22 heads, with different losses and outputs (labels, regression, descriptions, etc). Pengi can scale to N tasks, with a single loss and training procedure, and the same output format (free-form text). To check the benefit of our formulation and remove the effect of additional data, we train a new CLAP (CLAP*) on the same 3.4M pairs as Pengi. We attach this as Table 2 in PDF in the global response. Overall, Pengi's strong performance still holds.
>
> **Q5: Line 181: When computing log likelihoods, do the authors normalize ...  the overall score?**\
> Yes, we normalized based on the number of tokens for the log-likelihood method. We found that the number of tokens does affect the overall score, a larger number of tokens may decrease performance.
>
> **Q6: Sec. 5.5: Does the presented result show that ... in Tab. 9?**\
> The quality of the embeddings depends in good part on the training data. For example, adding more speech or music-related data usually correlates with improvement on relevant task. However, other components in Pengi are equally important, such as the audio encoder, text encoder, and LLM. They provide information, such as acoustic and text semantics and context. For instance, if our model never saw the word "dog" in the training data, but it saw "bark" and "animal", it can still associate and generate "dog barking" as a description.
>
> **Minor question 1, 3, 4 and Typos**\
> We thank the reviewer and will fix the issues and typos the reviewer pointed out.
>
> **MQ2: The authors say the text encoder is frozen ... encoder come from?**\
> This can be any off-the-shelf text encoder. We tested with text encoders from CLAP, BERT, T5, and CLIP and found little to no difference on downstream task performance.
>
> **MQ5: Line 271: Why does Pengi outperform human performance ... warranted.**\
> Humans have limitations inherent to how much information a participant can handle at once. In the case of ESC50, humans listen to the audio once, and have to remember the audio content, task description, and choose among 50 different classes. Moreover, listeners have different degrees of familiarity with prototypical content from different sound classes, whereas Pengi has been exposed to similar content during training. In a sense, Pengi is an expert listener, whereas the humans in the listening experiment were not.
>
> **MQ6: It would be useful to include the best results from Mei ... audio retrieval.**\
> The evaluation method for retrieval is different for contrastive and generative models, which makes a fair comparison difficult. We comment on this as well as the performance difference in Section 6 paragraph 1. As a follow-up, we will update our manuscript to explicitly mention these numbers in Table 6.
>
> **MQ7: Why does Pengi work better on some zero-shot tasks ... to the training data?**\
> This is due to multiple factors like training data, contrastive learning formulation, and having a common audio-text multimodal space.
>
> **L1: Does not support ASR, which is, arguably, one of the most important audio task.**\
> Pengi focuses on non-speech audio but we include some speech-related downstream tasks like speech emotion recognition to evaluate generalization. We didn't include speech audio and their transcripts, therefore Pengi does not support ASR. To the best of our knowledge, bridging speech, non-speech audio, and music in a single model is still an open problem.
>
> **L2: Unclear if the model can learn new tasks ... can learn new tasks by prompting.**\
> We have explored this direction in Appendix Section A, where we show that the user can use a fixed text prompt like “generate metadata” and then provide more information or ask follow-up questions. This enables the user to steer the conversation with additional prompts (Fig. 6), such as “the background is”, “mention forest.”, etc. However, we acknowledge that Pengi still falls short of VLMs like Flamingo in this aspect, and we plan to investigate this further in the future.
>
> **L3: How much does the LLM help? What is ... scratch with the current dataset?**\
> We appreciate the reviewer’s comment. Our default LM is GPT-2 (124M), but we also tested GPT2-XL (1.5B). A larger LM improved performance on the open-ended task of Audio Q&A, had mixed results on Audio Captioning, and no significant change on close-ended tasks. We attach the AQA results below.
> | LLM | Parameters | Audio Q&A|
> ------|------------|----------|
> | GPT2-base | 128M |  0.645   |
> | GPT2-XL   | 1.5B |  0.701   |

---

> > ### Comment · Reviewer_iz2N · 2023-08-14
> >
> > Thank you for addressing the comments. The results using CLAP trained using the same data is especially interesting.
> >
> > Clarification regarding tokenizer and mapping: Wouldn't using the same tokenizer as the LLM put the tokens in the same space at the LLM? Why does it need additional mapping to "understand" the tokens?

---

> > > ### Author Response · Authors · 2023-08-14
> > >
> > > We sincerely thank the reviewer for carefully reading our paper and rebuttal. For any additional clarifications, we are more than happy to address them.
> > > ***
> > > **Q1: Wouldn't using the same tokenizer as the LLM put the tokens in the same space at the LLM?**\
> > > Yes, using the same tokenizer as the LLM will put them in the same space as LLM. However, our text encoder is not the same as the LLM (GPT2), so it's not producing tokens in the same space.
> > >
> > > **Q2: Why does it need additional mapping to "understand" the tokens?**\
> > > Additional mapping is needed to understand the tokens for the case where we use a text encoder different than LLM (GPT2). If we remove the text encoder, additional mapping is not needed to understand tokens. Instead, the additional mapper helps the LLM produces different text output for different text prompts. For example, "this is sound of" should produce "dog barking" and "this emotion is" should produce "happy". This is out-of-domain for the frozen LLM and therefore requires a way to adapt the LLM for our data. We choose prefix-tuning as our choice of adaptation and introduce mapper m2. There are other ways to achieve this functionality i.e. instead of using mapper m2, one can use LORA updates or gated cross-attention on LLMs. We choose to keep the LLM completely frozen and instead tune the prefix using a mapper m2.

---

### Author Rebuttal · Authors · 2023-08-08

We sincerely thank all the reviewers for recognizing our contribution and providing constructive feedback, especially for acknowledging that **this paper presents a novel audio language model** (Reviewer vWxh, 4yQP), **Performance on audio captioning is solid.** (Reviewer sUJx), **One of the first approaches that incorporate LLMs to create a general purpose audio LM** (Reviewer iz2N, sUJx) and **the evaluation is pretty extensive. Authors compared the proposed model on multiple benchmarks against multiple models** (Reviewer o28f, Reviewer vWxh, Reviewer 4yQP).
***
We would like to re-emphasize the novelty and technical contributions of this work. We present a novel and unified model that can handle both open-ended and close-ended audio tasks without relying on external modules or fine-tuning. Our key contributions are: (1) Pengi, the first Audio Language Model (ALM) in the literature, (2) audio task templates inspired by Instruction Tuning, and (3) an extensive evaluation on 22 downstream tasks showing that our unified model can achieve competitive and even sota performance in several tasks.

We summarize the two main questions brought up by the reviewers and address them here: \
**Question 1: Why is an explicit mapping needed for input text? What's the purpose of m2 here and do you have experimental results to show using m2 is helpful ?**\
We need a mapping network $m_2$ to bring the output of the text encoder into the space of the LM. The text input is a sentence that is passed to the text encoder. The encoder outputs a sentence-level embedding that is passed to $m_2$, which outputs a sequence of embeddings that the LM can "understand". We conducted two experiments to evaluate the effect of omitting $m_2$ and/or the text encoder. We denote Exp A as Pengi without the text encoder and $m_2$ (input text directly to LM), and Exp B as Pengi without the text encoder​ but with $m_2$ (input text to $m_2$).​ In Exp A, we found that removing $m_2$ resulted in a loss of coherence between the input text prompt and the output text. For example, an input prompt about identifying an emotion class "the emotion is " resulted in random text output and thus random performance. In Exp B, we removed text encoder but retain $m_2$. We obtained slightly lower results than our proposed architecture with both components. We attached this as Table 1 in the PDF below.

**Question 2: Concern about using CLAP as baseline. Whether the performance gains are due to additional data or Pengi's ALM formulation?**\
We chose CLAP as the baseline because it is the only Zero-Shot model with a comprehensive evaluation (16 downstream tasks). The next best evaluation was only on 8 tasks. Thus, providing no evidence of performance across other domains like speech and music, which tend to be the most difficult. For the rest of the Tables, we compared against SoTA results even if it came from different models and learning methods. We compared against SoTa Zero-Shot models in Table 8, a subset of Table 3, for Sound Event Classification. Even against SoTa from supervised learning models in Tables 5 and 7 for Audio Q&A and Audio Captioning respectively. Table 9, against SSL, supervised and trained on speech audio models.

To enhance our baseline from Table 3 and answer whether the performance gains are due to additional data or Pengi's ALM formulation, we performed a new experiment. We trained a CLAP model (CLAP*) on the same amount of 3.4M audio-text pairs as Pengi. We observed Pengi's strong performance still holds. We attached this as Table 2 in the PDF.

---

### Comment · Area_Chair_3iD9 · 2023-08-18
**Please read rebuttals**

Dear reviewers, if you didn't already, please read the rebuttals ASAP and at least acknowledge them explicitly.

Best,
Area Chair

---

### Decision · Program_Chairs · 2023-09-21

**Decision:**

Accept (poster)

**Comment:**

The paper presents Pengi, an "audio language model" (ALM) which is a text language model that is conditioned on a both audio and text prefixes. The name is misleading, as the model doesn't generate audio and the language modeling over audio tokens is not central to the generation, it is a prefixing/conditioning mechanism. Pengi is compared to CLAP in various audio tasks, where is mostly beats it. There is no support for ASR, which admittedly is a special(ized) task. The contributions of the paper are relatively limited, but several reviewers see merit in unifying lots of audio tasks in a "prompting+LLM generation" way. The method is simple and the paper is easy to follow. The rebuttal convinced some reviewers of the experimental validation but not o28f, who had valid concerns. Overall, the paper is borderline (in term of scope and results), but being sound and SOTA on enough benchmarks with a single model, it deserves inclusion in NeurIPS.